# Participation of L-Lactate and Its Receptor HCAR1/GPR81 in Neurovisual Development

**DOI:** 10.3390/cells10071640

**Published:** 2021-06-30

**Authors:** Samuel Laroche, Aurélie Stil, Philippe Germain, Hosni Cherif, Sylvain Chemtob, Jean-François Bouchard

**Affiliations:** 1Neuropharmacology Laboratory, School of Optometry, Université de Montréal, Montreal, QC H3T 1P1, Canada; samuel.laroche.1@umontreal.ca (S.L.); aurelie.stil@umontreal.ca (A.S.); philippe.germain.1@umontreal.ca (P.G.); hosni.cherif@hotmail.com (H.C.); 2Department of Pediatrics, Research Center-CHU Sainte-Justine, Montreal, QC H3T 1C5, Canada; sylvain.chemtob@umontreal.ca; 3Department of Ophtalmology, Faculty of Medicine, Université de Montréal, Montreal, QC H3T 1J4, Canada

**Keywords:** lactate, GPR81, HCAR1, retinal ganglion cells, growth cone, dLGN, retina, axon, 3,5-DHBA

## Abstract

During the development of the retina and the nervous system, high levels of energy are required by the axons of retinal ganglion cells (RGCs) to grow towards their brain targets. This energy demand leads to an increase of glycolysis and L-lactate concentrations in the retina. L-lactate is known to be the endogenous ligand of the GPR81 receptor. However, the role of L-lactate and its receptor in the development of the nervous system has not been studied in depth. In the present study, we used immunohistochemistry to show that GPR81 is localized in different retinal layers during development, but is predominantly expressed in the RGC of the adult rodent. Treatment of retinal explants with L-lactate or the exogenous GPR81 agonist 3,5-DHBA altered RGC growth cone (GC) morphology (increasing in size and number of filopodia) and promoted RGC axon growth. These GPR81-mediated modifications of GC morphology and axon growth were mediated by protein kinases A and C, but were absent in explants from *gpr81^−/−^* transgenic mice. Living *gpr81^−/−^* mice showed a decrease in ipsilateral projections of RGCs to the dorsal lateral geniculate nucleus (dLGN). In conclusion, present results suggest that L-lactate and its receptor GPR81 play an important role in the development of the visual nervous system.

## 1. Introduction

The physiological significance of lactic acid and its conjugate base lactate have been a major source of controversy since their discovery in biological tissues. Lactic acid was long considered to be simply the waste product of anaerobic glycolysis. Mainly occurring as the L-enantiomer in physiological conditions, lactate is now known to have multiple effects on cell homeostasis, serving as a metabolic fuel and buffering agent, while also acting as a signaling molecule, also known as “Lactormone”. This signaling action is obtained via the hydroxycarboxylic acid receptor 1 (HCAR1) [1,2]. Also known as GPR81, this is a G-protein-coupled receptor activated by L-lactate and the exogenous agonist 3,5-dihydroxybenzoic acid (3,5-DHBA) [3]. GPR81 is expressed in diverse organs, including adipose tissues, skeletal muscle, liver, kidney, brain, and retina [4,5].

The retina and central nervous system have an inherently high energy demand due to the continuous depolarization of neuronal membranes [6]. During the development of the visual system, the axons of the retinal ganglion cells (RGCs) that form the optic nerves follow chemotropic molecules to grow toward and across the optic chiasma [7]. In mammals, most of the axons cross at the optic chiasm to reach the contralateral side of the optic tract while only axons from RGCs on the ventro-temporal side of the retina do not cross at the optic chiasm. Ventro-temporal RGCs specifically express the ephrin receptor (EphB1) on their GC. When the GCs come into contact with the Ephrin-B2 ligand secreted by radial glial cells in the midline of the optic chiasm, a chemorepulsive effect redirects them and the axons project to the ipsilateral optic tract [8]. Following the optic tract, some of RGC axons join the dorso-lateral geniculate nucleus (dLGN) in the thalamus, making synapses with dLGN neurons. These neurons project their axons to the layer 4 of the primary visual cortex [9].

The consequent high metabolic demand for ATP production is met by glycolysis, resulting in L-lactate generation, despite a rich endowment of mitochondria for aerobic respiration. ATP generation by oxidative phosphorylation is limited and high energy demand leads to an increase of glycolysis filling the ATP equilibrium [2,10]. It has been previously reported that the lactate dehydrogenase (LDH-1) subunit, which preferentially catalyzes the conversion of L-lactate to pyruvate, is found in neurons and astrocytes [11,12]. However, LDH-5, which is more highly expressed in astrocytes, preferentially favors the conversion of pyruvate back to L-lactate [11]; LDH-5 is also present in glycolytic tissues such as skeletal muscles [13,14]. This cell-type distribution of LDH subunits indicates a close link with characteristic features of L-lactate metabolism, which has implications for neuronal development, growth, and survival [6,11].

GPR81 has been reported to mediate effects of L-lactate in diverse processes, including wound healing, angiogenesis, neuroprotection, cancer cell survival, attenuation of inflammation, and antilipolytic effects [5,15,16,17,18,19,20]. Recent studies have revealed the presence of GPR81 in Müller cells and retinal ganglion cells (RGCs) [21]. Its activation in these cells regulates angiogenic Wnt ligands and Norrin, which together participate in intra-retinal vascularization. Receptors specific for other metabolic intermediates have also been shown to govern angiogenesis and neuronal growth in the visual system. Thus, receptors for the Krebs cycle intermediates succinate (GPR91) and α-ketoglutarate (GPR99) control vascular and axonal growth [22,23]. Hence, there is a close link between metabolism and cell signaling in regulating the development of vascular and neuronal systems. However, the role of L-lactate and its cell membrane target GPR81 during the development of the central nervous system (CNS) is yet to be explored. In this study, we built upon established findings in the CNS to investigate the role of GPR81 during development of the visual system. More specifically, we investigated the effects of L-lactate/GPR81 on axonal growth guidance in the developing mouse neuro-visual system.

## 2. Materials and Methods

### 2.1. Ethics Statement and Animals

All animal procedures were performed in accordance with the Animal Care Committee of the University of Montreal following the guidelines from the Canadian Council on Animal Care. The *gpr81*^−/−^ mice were purchased from Lexicon Pharmaceuticals (The Woodlands, TX, USA). These mice were developed by the insertion of a 4-kb IRES-lacZ-neo cassette in the trans-membrane domain 2 coding region (100 base pairs) of the *gpr81* gene on C57BL/6J mice [20], with control studies in C57BL/6J WT control mice. Other experiments were undertaken in golden Syrian hamsters (Charles River Laboratories, Saint-Constant, QC, Canada). All animals were maintained in an environmentally controlled room held at 21 ± 2 °C, and with 12 h dark/light circle. Mice and hamsters of both sexes were used in this study. Food and water were provided ad libitum.

### 2.2. Genotypic Screening

Mice were genotyped by PCR using the amputated tail tip for DNA extraction. Tail samples were immersed in 50 mM NaOH, incubated for 20 min at 95 °C, vortexed, and neutralized with 1 M Tris-HCl, pH 8.0. The samples were then revortexed and centrifuged for eight minutes at 13,000 rpm in a Fisher Scientific^TM^ accuSpin^TM^ Micro R centrifuge. The supernatant was taken for DNA amplification and added to the PCR reagent mix containing PCR Buffer, MgCl_2_, dNTP mix, Taq DNA polymerase, forward and reverse primers. PCR cycle conditions were: 5 min at 95.0 °C, 30 cycles of three steps (1 min at 50.0 °C, 1 min at 72.0 °C and 1 min at 95.0 °C). Genotype of the *Gpr81^−/−^* mice was confirmed using specific primers for wild-type (WT), with normal mouse DNA as a control. The primer pairs w (forward: 5′-CATCTTGTTCTGCTCGGTCA–3′ and reverse: 5′-GAGGAAGTAGAGCCTAGCCA-3′) were used to amplify a 160 bp fragment present in the *gpr81*^+/+^ genome but absent in the *gpr81*^−/−^ mice.

### 2.3. Reagents

L-(+)-lactic acid, bovine serum albumin (BSA), ciliary neurotrophic factor (CNTF), DNase, forskolin, insulin, laminin, poly-D-lysine, progesterone, selenium, putrescine, gelatin from porcine skin, chromium(III) potassium sulfate dodecahydrate, sucrose, sodium chloride (NaCl), PCR reagent mix, potassium chloride (KCl), hydrochloric acid (HCl), disodium hydrogen phosphate (Na_2_HPO_4_), potassium phosphate monobasic (KH_2_PO_4_), *gpr81* primers, mouse anti-Brn-3a (MAB1585), rabbit anti-GPR81-S296 (SAB1300790), mouse anti-MAP2 (M9942), transferrin, trypsin, and triiodothyronine were purchased from Sigma Aldrich (Oakville, ON, Canada). B27, N2, fetal bovine serum (FBS), neurobasal medium (NB), penicillin-streptomycin, Minimum Essential Medium Eagle medium, Spinner Modification (S-MEM), and sodium pyruvate were purchased from Life Technologies (Burlington, ON, Canada). The standard goat serum, Peroxidase-AffiniPure Donkey anti-rabbit IgG, and Peroxidase-AffiniPure Donkey anti-mouse IgG were from Jackson ImmunoResearch (West Grove, PA, USA). N-acetyl cysteine (NAC) was acquired from EMD technologies (Saint-Eustache, QC, Canada). 3,5-DHBA was purchased from Tocris (Oakville, ON, Canada). Cholera toxin subunit B (CTb) recombinant conjugate with Alexa Fluor 555 (C34776) and 647 (C34778), GlutaMAX™ Supplement, green Neg-50 Frozen section medium, Hank’s 1X Balanced Salt Solutions (HBSS), Tween 20, Triton X-100, paraformaldehyde (PFA), sucrose, TEMED, sodium hydroxide (NaOH), Tris base, Taq DNA Polymerase, Shandon Immu-Mount, AlexaFluor donkey anti-mouse 488, Alexa Fluor 488 goat anti-rabbit IgG (H+L), AlexaFluor goat anti-rabbit 546, Alexa Fluor 546 goat anti-mouse, and Alexa Fluor 546 phalloidin were obtained from Fisher Scientific (Ottawa, ON, Canada). Rabbit anti-PKA C-α (#4782), rabbit anti-Phospho-PKA C (Thr197) (#4781), rabbit anti-PKCα (#2056), and rabbit anti-Phospho-PKC (pan) (βII Ser660) (#9371) were acquired from Cell Signalling Technology (Whitby, ON, Canada). Heparin and sterile saline solution (0.9%) were purchased from CDMV (St-Hyacinthe, QC, Canada).

### 2.4. Tissue Preparation for Immunohistochemistry

Adult mice and postnatal day 5 (P5) golden Syrian hamsters were euthanized by an isoflurane overdose. A transcardial perfusion was conducted with 10 U/mL of heparin in 60 mL of phosphate-buffered 0.9% saline (PBS; 0.1 M, 4 °C, pH 7.4), followed by 60 mL phosphate-buffered 4% paraformaldehyde 4 °C (PFA). Following the harvesting of mouse embryos E14-16 extraction by Cesarian, they were deeply anesthetized by hypothermia. The orbits were removed and two small holes were made in the cornea prior to immersion fixation in 4% PFA 4 °C for 60 min. The eyecups were washed in PBS, cryoprotected in 30% sucrose overnight, embedded in Neg 50 medium, flash-frozen, and kept at −80 °C until processing. Sections 14 μm-thick were cut with a cryostat (Leica Microsystems, Exton, PA, USA) and mounted on slides coated with gelatin/chromium (double-frosted microscope slides, Fisher Scientific, Ottawa, ON, Canada).

### 2.5. Immunohistochemistry

Frozen sections were thawed, washed three times for five minutes each time in 10 mM PBS with 0.05% Tween 20(PBST), and then blocked in 1% BSA, gelatin, and 0.5% Triton X-100 in PBS for one hour. The sections were then co-incubated overnight with rabbit anti-GPR81 and mouse anti-Brn3a (a specific marker for RGCs). The next morning, sections were washed three times for five each time minutes in PBST and incubated for one h with secondary antibodies: AlexaFluor donkey anti-mouse 488 and AlexaFluor goat anti-rabbit 546. After three washes in PBST, the sections were slide mounted with Shandon Immu-Mount.

### 2.6. L-Lactate Solution

Daily fresh solution of 100 mM L-lactate was made using 0.1 g of L-(+)-lactic acid (MW = 90.08 g/mol) dissolved in 10 mL of NB (vehicle). The pH was adjusted to 7.4 ± 0.1 with NaOH, the volume reached 11 mL with NB, and the solution was filtered with Corning™ PES 0.20 µm pore Syringe Filters (09-754-29). The prepared L-lactate solution was equilibrated for at least one h at 37 °C in a 5% CO_2_ incubator before use in experiments in vitro.

### 2.7. Retinal Explant Culture

The retina were isolated from E14-E16 mouse embryos, dissected into small segments in HBSS, and plated on 12 mm diameter glass coverslips previously coated with poly-D-Lysine (20 μg/mL) and laminin (5 μg/mL) placed in 24-well plates. The explants were cultured in NB medium supplemented with 100 U/mL penicillin/streptomycin, 5 μg/mL NAC, 1% B27, 40 ng/mL selenium, 16 μg/mL putrescine, 0.04 ng/mL triiodothyronine, 100 μg/mL transferrin, 60 ng/mL progesterone, 100 μg/mL BSA, 1 mM sodium pyruvate, 2 mM glutamine (glutaMAX^TM^), 10 ng/mL CNTF, 5 μg/mL insulin, and 10 μM forskolin at 37 °C and 5% CO_2_. At 0 DIV, starting 1 h following plating, the explants were treated for 15 h for projection analysis or for 1 h at one day in vitro (DIV) for the growth cone morphology analysis. Photomicrographs were taken with a Olympus IX71 microscope (Olympus, Markham, ON, Canada) and the axonal projection and growth cone measurement analysis were made using ImageJ software. These analyses were performed by operators blind to the experimental condition.

### 2.8. Growth Cone Behavior Assay

Similarly, embryonic retinal explants were cultured in Thermo Scientific™ Nunc™ Lab-Tek™ Chambered Coverglass with a borosilicate glass bottom (Lab-Tek; Rochester, NY, USA). After one or two DIV, explants were transferred to an incubator (Live cell chamber) at 37 °C and 5% CO_2_, mounted on an inverted Olympus IX71 microscope (Olympus, Markham, ON, Canada). A micropipette was positioned at a 45° angle about 100 µm from the growth cone of interest, as previously described [24,25,26]. A micro-injector (Harvard Apparatus, St-Laurent, QC, Canada) was used to deliver NB vehicle or L-lactate 20 mM (pH 7.4) at a rate of 0.1 µL/min of in the NB. Measurements were performed with ImageJ software at baseline and after 60 min of these treatments.

### 2.9. Primary Neuron Culture

Primary cortical neurons were used in this study because of the ease in culturing and harvesting sufficient numbers for biochemical assays, which is technically difficult for RGCs. C57BL/6J WT pregnant mice were used to obtain E14-16 embryo brains. The superior layer of each cerebral cortex was isolated and transferred to 2 mL S-MEM containing 2.5% trypsin and 2 mg/mL DNase and incubated at 37 °C for 15 min. After trituration, the pellet was transferred to 10 mL S-MEM with 10% FBS and stored at 4 °C. The pellet was again transferred in 2 mL S-MEM supplemented with 10% FBS and triturated three or four times. The supernatant was transferred to 10 mL NB medium. Dissociated neurons were counted under a microscope and plated at a density of 100,000 cells per well on 12 mm glass coverslips previously coated with poly-D-lysine (20 μg/mL) for immunocytochemistry, or at 250,000 cells per 35 mm Petri dish for western blot analysis. Neurons were cultured for two-four days in NB medium supplemented with 1% B-27, 100 U/mL penicillin/streptomycin, 0.25% N2, and 0.5 mM of glutaMAX^TM^. They were then treated with the GPR81 agonist L-lactate for 1 h to study acute effects on growth cone morphology, or for various other intervals for identifying the activation of signaling pathways by western blot analysis.

### 2.10. Immunocytochemistry

After treatment, retinal explants and primary cortical neuron cultures were washed with PBS (pH 7.4), fixed in 4% PFA (pH 7.4), and blocked with 2% normal goat serum (NGS) and 2% BSA in PBS containing 0.1% Tween 20 (pH 7.4) for 30 min at room temperature. The samples were then incubated overnight at 4 °C in a blocking solution containing anti-GPR81, anti-MAP2. The following day, the samples were washed and labeled with Alexa Fluor 488 and 546 secondary antibodies against the host species of the primary antibodies, Hoechst 33258, or AlexaFluor Phalloidin 546. The coverslips were mounted with Shandon Immu-Mount and photomicrographs were taken with an Olympus IX71 microscope (Olympus, Markham, ON, Canada) for quantitative analysis on ImageJ software.

### 2.11. Western Blot Analysis

Following L-lactate treatment at various time points, the primary cortical neurons were washed with cold 4 °C PBS (pH 7.4) and then lysed with 125 µL of laemmli sample buffer pre-warmed to 100 °C. The samples were then frozen at −20 °C until the day of Western blot analysis. On that day, the samples were thawed at 4 °C, placed in a 100 °C dry bath for 10 minutes, quickly vortexed and then centrifuged at 13,000 rpm at 4 °C. Twenty microliter samples were then resolved on a 10% SDS-polyacrylamide gel along with the trihalo compound from TGX Stain-Free Technology (Bio Rad, Mississauga, ON, Canada). During the electrophoresis, the trihalo compound covalently modifies tryptophan residues in the proteins to impart a fluorescence signal. Visualization of this signal was obtained by UV excitation in a Chemidoc imaging system (Bio-Rad). After this activation, gels were transferred onto a PVDF membrane with a TransBlot Turbo (Bio-Rad), and blots were imaged on the Chemidoc to reveal the total protein transferred on the membrane, which was used for later normalization of antibody signals to total protein. The blots were then blocked with 2% BSA in TBST (Tris 10 mM and NaCl 150 mM saline with 0.1% Tween 20) for 1 h and incubated overnight with antibodies against Phospho-PKA, PKA, Phospho-PKC, and PKC. They were then exposed to the species-appropriate HRP-coupled secondary antibodies for two h in blocking buffer, and dected using the Chemidoc with Clarity Max ECL substrate from Bio-Rad. The target protein expressions were then analyzed on the Image Lab v.6.0.1 software. All procedures were completed according to Bio-Rad protocols [27].

### 2.12. Eye Specific Segregation

In this in vivo experiment, *gpr81^+/+^* and *gpr81^−/−^* adult mice under anesthesia received an intraocular injection of 2 µL of 1% (mg/mL) CTb in 0.9% sterile saline conjugated to AlexaFluor 555 into the left vitrious humor eye, and with CTb conjugated to AlexaFluor 647 into the right vitrious humor eye. Four days after the injections, the animals were anesthetized and perfused transcardially, as described in the tissue preparation section above. The brains were removed, post-fixed in PFA 4% overnight, and then cryoprotected by immersion in gradient sucrose solutions of 10%, 20%, and 30% until the brains sank, followed by storage at −80 °C. Coronal sections 40 µm thick were cut on glass slides in a cryostat, air-dried, and mounted with Shandon Immu-Mount. Fluorescent images of entire brain sections were taken using 561 and 640 nm laser emissions and a 4× objective on an Fluoview FV3000 Olympus Confocal microscope to identify the three sections containing the largest ipsilateral projection. The dorsal lateral geniculate nucleus (dLGN) was scanned in these sections with a 20× objective and 2× amplification (total magnification of 40×). Z-stacks of 19 images with both lasers were taken from top to bottom of the signal emissions. The colocalization of both channels with a multi-threshold analysis was performed on CellSens Dimension software. The percentage of ipsilateral signal overlapping with the contralateral signal was measured for each stack. The mean for the 19 stacks in each section was reported for each threshold, and differences evaluated by two-way ANOVA with Tukey post hoc testing [28,29].

### 2.13. Statistical Analysis

Data were imported in GraphPad Prism 8 software. Tests for normal distribution were performed by an Anderson–Darling test (α = 0.05). Depending on parametric or nonparametric distribution, the appropriate statistical analysis was computed. Values were reported as the mean ± SEM.

## 3. Results

### 3.1. GPR81 Is Expressed in the Retina

We used rabbit anti-GPR81-S296 (SAB1300790) immunohistochemistry to identify the retinal laminar distribution of GPR81. Results in retina from P5 Syrian gold hamster pups, E16 mouse embryos, and adult mice all showed retinal GPR81 expression. GPR81 protein immunoreactivity was consistently detected in the Outer Nuclear Layer (ONL), Inner Nuclear Layer (INL), Inner Plexiform Layer (IPL), RGCs, and the RGC fiber layer of hamsters (Figure 1A–C). GPR81 immunoreactivity was detected in all layers of the embryonic and adult mouse retina (Figure 1D-I). The colocalization of GPR81 with the specific RGC marker Brn-3a showed that GPR81 is predominantly expressed in RGCs and in the RGC fiber layer. From this result, we inferred possible involvement of GPR81 in retinal development and in projection navigation towards brain innervation targets.

Primary cortical neurons and retinal explant cultures in vitro also expressed the lactate receptor; GPR81 fluorescence signal was visualized on the neuronal soma, neurites, GC, and filopodia (Figure 2A, Appendix A).

### 3.2. GPR81 Influences GC Morphology and Axon Growth

A number of GPCRs exert effects on axon guidance [22,24,25,26,30,31]. To determine the implication of GPR81 in regulating GC morphology, we cultivated embryonic retinal explants during 1 DIV. Afterwards, treatment for one h with the GPR81 agonist L-lactate (10 mM) significantly increased the RGC GC surface area by 44.2 ± 9.6%, whereas 3,5-DHBA (300 µM) increased surface area by 56.1 ± 12.0% compared to the vehicle control. These same treatments increased filopodia numbers of GC by 38.1 ± 8.1% and 31.0 ± 6.5%, respectively on retinal explants obtained from *gpr81*^+/+^ mice. The corresponding changes induced by GPR81 agonists were abrogated in similar retinal explants obtained from *gpr81*^−/−^ mouse embryos (Figure 2B–C).

We next measured the effect of GPR81 agonists on axon growth of RGCs on retinal explants. Interestingly, RGCs from *gpr81^+/+^* mice incubated with GPR81 agonists for 15 h increased axon outgrowth by 40.7 ± 2.2% (10 mM L-lactate) and 38.7 ± 4.2% (300 µM 3,5-DHBA) (Figure 2D–E). As expected, RGCs obtained from *gpr81*^−/−^ mice did not respond to the agonist treatment (−4.2 ± 3.7% and 4.8 ± 5.0%, respectively) (Figure 2D–E). This impact of L-lactate on GC morphology and axon growth was confirmed with time-lapse live cell imaging in the GC behavior assay. Here, we exposed a GC of a retinal explant to a microgradient of L-lactate concentration to visualize any axon growth or guidance effects. In this experiment, we measured the GC surface area, filopodia numbers, axon length, and the angle between the GC and the micropipette at baseline (T0) and at 60 min of drug exposure (T60). The time-dependent differences were obtained by subtraction. At 60 min, the vehicle-treated explants (NB medium) showed negligible changes in the GC (−0.42 ± 3.03 µm^2^ area), 0.5 ± 0.5 filopodia, and −1.39 ± 2.78 µm axon growth; in contrast, L-lactate-treated explants displayed mean increases of 22.8 ± 11.1 µm^2^ GC area, 4.0 ± 1.8 in number of filopodia, and 4.31 ± 2.14 µm in axon growth. However, we did not observe any significant changes in the angle of orientation after the addition of lactate (−2.1 ± 2.0° for NB medium versus 2.5 ± 3.5° for 20 mM L-lactate gradient; Figure 3).

### 3.3. Lactate Increases PKC and PKA Phosphorylation

GPR81 is coupled to G_i_, such that agonism induces Ca^2+^ release in CHO-K1 adipocytes, resulting in decreased lipolysis [32]. To determine the signaling pathway resulting in GPR81 activation, we treated primary neurons obtained from mouse embryos with a GPR81 agonist (10 mM L-lactate). Interestingly, there was an increase of PKC phosphorylation (2.1 ± 0.4 fold) after five minutes and an increase of PKA phosphorylation (1.9 ± 0.2 fold) at 15 min after the addition of 10 mM L-lactate (Figure 4A). These data show that GPR81 agonism stimulates the PKC and the PKA signaling pathways in primary neurons.

### 3.4. GPR81 Affects Retinothalamic Projections In Vivo

We next studied the contribution of GPR81 to the development of retinothalamic projections in living transgenic *gpr81^−/−^* mice. Left and right RGC projections were visualized with the neuron tracer CTb conjugated to Alexa fluorescent molecules. RGC projections normally segregate through various brain structures to join dLGN [33]. Interestingly, adult *gpr81^−/−^* mice had significantly fewer ipsilateral projections in the dLGN, irrespective of the intensity threshold used for the analysis (Figure 5). This observation demonstrates, for the first time, an essential role of GPR81 in the normal development of the rodent retinothalamic pathway.

## 4. Discussion

Lactic acid and L-lactate have drawn considerable attention since the early days of metabolic research. Indeed, the indisputable importance of lactate in the glycolytic pathway may have led to persisting misconceptions about the bodily functions of L-lactate [1]. The introduction of the astrocyte-neuron Lactate shuttle hypothesis (ANLSH) [34] led to a more mature understanding of the importance of L-lactate metabolism in the CNS. According to this model, glucose is predominantly taken up by astrocytes during neuronal activation. The astrocytes produce and release L-lactate, which serves as the primary metabolic fuel for activated neurons [35]. Furthermore, glucose consumption by the outer retina forms L-lactate by aerobic glycolysis [36,37], in metabolic support of RGCs. The recent discovery of receptors for metabolic intermediates has brought new physiologic perspectives on the relevance of L-lactate concentrations in tissue. We undertook in this study to explore the localization and neural functions of GPR81 in the developing retina and its CNS projections. We show for the first time expression of the lactate receptor GPR81 in the retina of hamsters and mice, noting its high abundance in the RGC layer and in the RGC nerve fiber layer. This result was confirmed by studies in vitro using retinal explants and cortical neurons of embryonic mice, which both proved to express GPR81 on their soma, axons, GCs, and GC filopodia. We demonstrated that specific activation of the GPR81 receptor by its agonists modulates GC size, filopodia numbers, and axon growth of RGCs. In vitro, treatment with L-lactate at a physiological concentration increased phosphorylation of PKC and PKA, which are key mediators of the pathways that likely promote neurite growth effects [38]. Importantly, genetic absence of GPR81 curtailed formation of RGC projections to the dLGN, as evident in the *gpr81^−/−^* mouse in vivo. Altogether, these results suggest an essential involvement of L-lactate and its receptor GPR81 in the development of RGCs and their axon projections in the CNS.

Previous findings for GPR81 impart an extra layer of complexity with regard to its involvement in CNS development, especially for retinal projections. The colocalization of GPR81 with glutamine synthetase, a specific marker of Müller cells, was previously established in early postnatal mice (P8-17). GPR81 activation in Müller cells regulates Norrin and Wnt ligands, which in turn promote angiogenesis and produce stimulate production of neuroprotective growth factors in the retina [21]. In our present embryonic model (E15), Müller cells have not yet been differentiated in the developing retina. Accordingly, the mechanism responsible for axon growth at this development stage likely differs from the retinal angiogenic effects produced by Müller cells in postnatal rodents [39].

L-lactate was long considered to be a metabolic waste of anaerobic glycolysis, but it is now known to be a metabolic fuel of aerobic glycolysis, especially in neurons [1,2]. In this capacity, L-lactate is converted by LDH to pyruvate, which then enters into the Krebs cycle for ATP generation in mitochondria [40]. In order to confirm that the present findings of L-lactate effect are not due trivially to an increase in the energy metabolite pool, we used 3,5-DHBA as a non-metabolized specific GPR81 agonist [3]. Activation of GPR81 by L-lactate and 3,5-DHBA of RGCs in retinal explants from embryonic mice positively modulated GC morphology and increased axon projection length. Interestingly, these effects of GPR81 agonist treatment were absent in explants from *gpr81^−/−^* mice. These findings support our claim that L-lactate and 3,5-DHBA act on RGCs through GPR81 agonism.

RGCs in the intact organism project their axons from retina to transmit visual information to CNS targets. GCs must detect and respond to a complex combination of chemotactic signals to direct their axons through the visual pathway. Our time-lapse microscopy experiments confirmed that L-lactate concentration gradients modify GC morphology and increase axon growth without having a significant effect on axon guidance *per se*. However, L-lactate was without effect in explants from retinal mice genetically depleted of GPR81, thus suggesting a crucial role for GPR81 in the retinothalamic pathway formation.

We show in this study that L-lactate activates PKC and PKA through agonism at GPR81, as was reported previously. Indeed, PKC activation is well-known to participate in synaptic remodeling, presynaptic plasticity, and neuronal repair [41,42]. Some PKC isoforms are known to influence neurite adhesion and outgrowth in various neuronal cell types, including RGCs [43,44]. Activation of these second messenger pathways results in a reorganization of the growth cone cytoskeleton, which manifest to microscopic examination of the filopodia, lamellipodia, and axonal growth in vitro [45,46,47]. Our present results are thus also in accordance with studies showing that PKA activation increases GC area and filopodia numbers to promote RGC axon extension during development [48].

In conclusion, as demonstrated in this study for the first time, the GPR81 L-lactate receptors contribute critically to neuro-visual development in the rodent retina and CNS. This study thus provides a foundation for the ongoing investigation of these actors in broader aspects of central and peripheral nervous system development. Present results support the formulation of new hypotheses for elaborating effective therapies targeting the development and regeneration of the nervous system. In fact, our results suggest a possible therapeutic role of L-lactate and other GPR81 agonists in the neuroregenerative and neuroreparative field. First of all, L-lactate can act by itself as an efficient energy substrate [49]. It can also protect and attenuate neuronal death induces by oxygen and glucose deprivation following cerebral ischemia [50,51]. In traumatic brain injury model, preconditioning with L-lactate promotes plasticity-related expression helping to reduce neurological deficits effect via the GPR81 signaling pathway [52]. In accordance with our findings, the utilization of L-lactate or a GPR81 agonist could stimulate axon regeneration and prevent neuronal degeneration.

## Figures and Tables

**Figure 1 cells-10-01640-f001:**
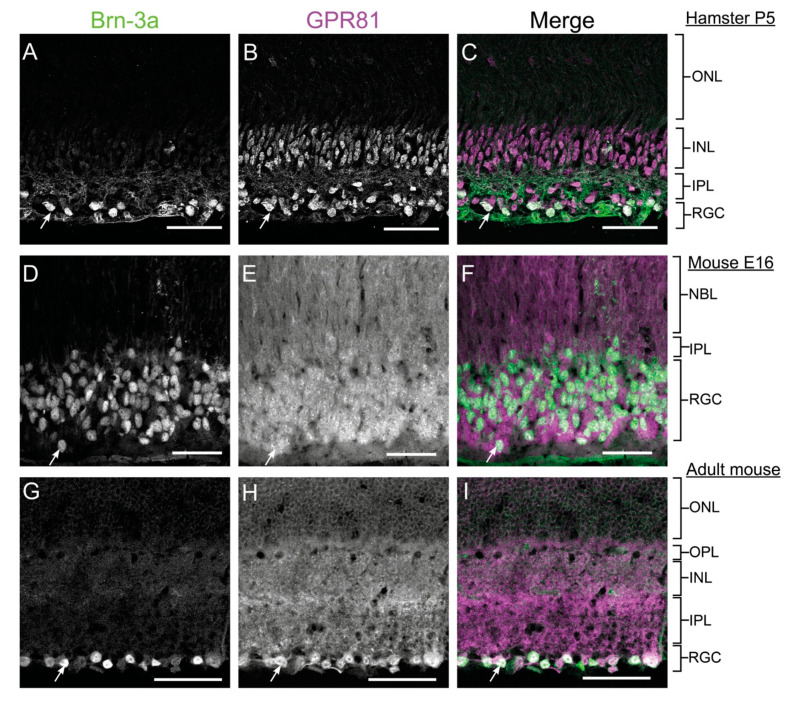
GPR81 is Expressed in the Retina. Expression of GPR81 in RGCs of retinal sections from P5 hamster pups (**A**–**C**) E16 WT mouse embryo (**D**–**F**), and adult mouse (**G**–**I**). ONL: outer nuclear layer, RGC: retinal ganglion cells, OPL: outer plexiform layer, INL: inner nuclear layer, NBL: neuroblast layer, IPL: inner plexiform layer. The white arrows indicate colocalization of GPR81 and Brn-3a. Scale bars: 50 µm.

**Figure 2 cells-10-01640-f002:**
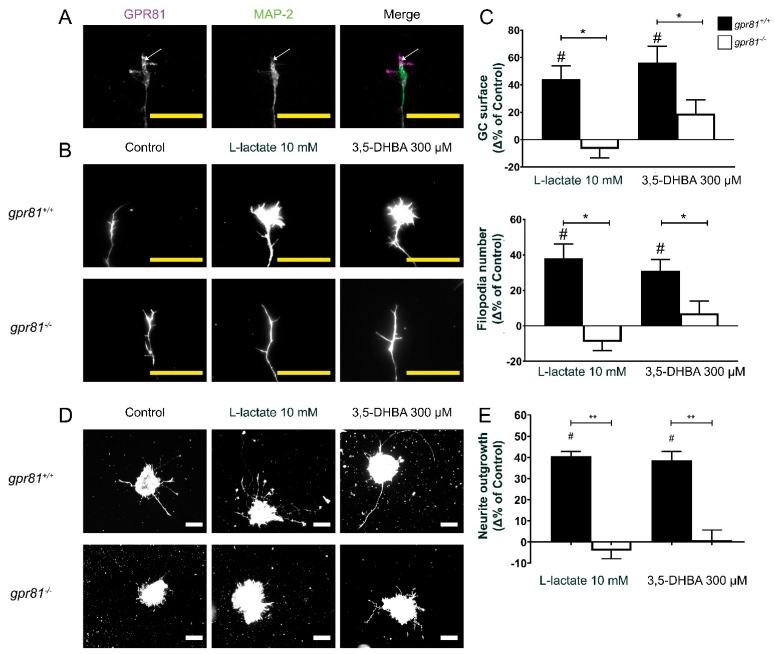
L-lactate and 3-5-DHBA influence GC dynamics and increase axon growth via GPR81 agonism. (**A**) GPR81 is expressed in GCs in vitro. (**B**) Microscopic photography of GCs following 1 h treatment with GPR81 agonists. Explants were marked with phalloidin conjugated with Alexa fluor 546. (**C**) Analysis of GC surface area and filopodia number (*n* = 48–147 GCs per condition). Values are presented as the means ± SEM. (**D**) Microscopic photography of retinal explants following 15 h treatment with the GPR81 agonists. Explants were marked with phalloidin conjugated with Alexa fluor 546. (**E**) Analysis of RGC axon projections (*n* = 278–2572 axons per condition). Values are presented as the means ± SEM. ^#^ Indicates significant changes compared to the control group. * Indicates significant changes *p* < 0.05 between *gpr81^+/+^* and *gpr81^−/−^* genotype. ** Indicates significant changes *p* < 0.01 between *gpr81^+/+^* and *gpr81^−/−^* genotype. Statistical test used is the Krustal–Wallis nonparametric ANOVA. White scale bar: 100 µm. Yellow scale bar 25 µm.

**Figure 3 cells-10-01640-f003:**
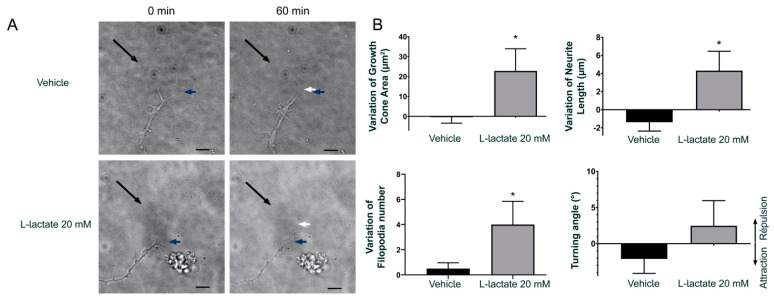
L-lactate influences GC morphology and axon growth, but not turning. (**A**) Time-lapse microscopy time of a DIV1-2 mouse RGC growth cone before and at 60 min after the addition of L-lactate. The black arrows indicate the microgradient direction created by a micropipette with a tip of 2 µm diameter and positioned at a 45° angle and around 100 µm distal to the growth cone. The blue arrows indicate the tips of the growth cone before the application of L-lactate or vehicle (T0). The white arrows indicate the position of the growth cone 60 min later (T60). (**B**) Analysis of growth cone dynamics. The values of the growth cone area, filopodia number on the growth cone, axon length, and the angle of the growth within the L-lactate microgradient were measured before (T0) and at 60 min (T60) after the addition of L-lactate or vehicle. Bar graphs show the means ± SEM of the effects induced by the treatments (*n* = 6–8 cells per condition). * Indicates significant changes compared to the vehicle group, compared by parametric unpaired *t*-test (*p* < 0.05). Scale bar = 15 µm.

**Figure 4 cells-10-01640-f004:**
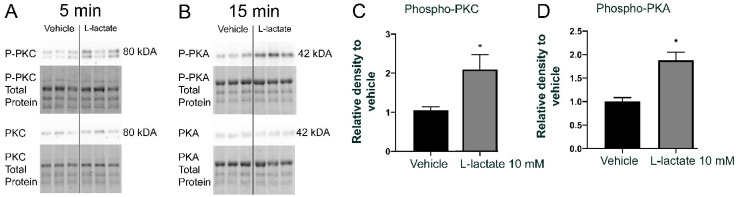
L-lactate increases PKC and PKA phosphorylation (**A**,**B**) Protein expression levels of P-PKC, PKC, P-PKA, PKA, and total proteins in primary neuron cultures incubated with L-lactate (10 mM) at 37 °C for five min (for PKC) and 15 min (for PKA). The total protein Western blots confirmed equal loading in all lanes with TGX Stain-Free Technology. (**C**,**D**) Values are presented as the means ± SEM. * Indicates significant changes compared to the control group, as compared by the nonparametric ANOVA Krustal Wallis (*p* < 0.05) with Dunn’s post hoc correction.

**Figure 5 cells-10-01640-f005:**
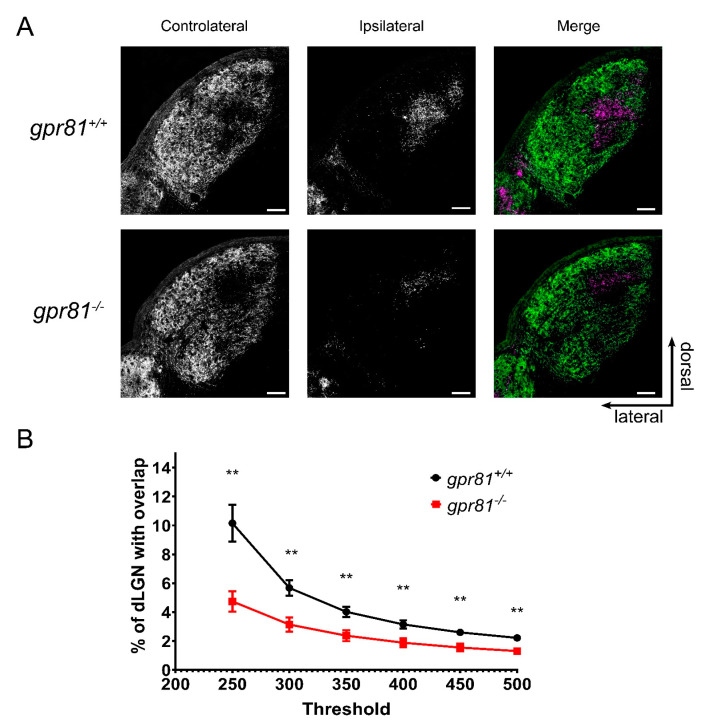
GPR81 Affects Retinothalamic Projections in vivo. (**A**) Micrographs of the dorsal lateral geniculate nucleus (dLGN) following the injection of CTb-Alexa 555 in the left eye and CTb-Alexa 647 in the right eye of *gpr81^+/+^* and *gpr81^−/−^* mice. Controlateral and ipsilateral projections from both eyes RGC segregate in the dLGN. RGC projections from both eyes are shown in the merged images. (**B**) Line graph shows the percentage of ipsilateral projections overlapping in the contralateral projections expressed as the mean ± SEM. Percentage of overlapping signals is significantly lower in *gpr81^−/−^* mice irrespective of the intensity analysis threshold. ** Indicates significant changes between *gpr81^+/+^* (*n* = 18) and *gpr81^−/−^* (*n* = 12) genotypes, according to two-way ANOVA with Tukey post hoc test (*p* < 0.01). Scale bars: 100 µm.

## Data Availability

All relevant data are within the paper and its Appendix A.

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
