# Peer review of "Participation of L-Lactate and Its Receptor HCAR1/GPR81 in Neurovisual Development"

_cells, 2021, doi:10.3390/cells10071640_

Round 1
Reviewer 1 Report
The paper is very clear, well structured and written in good English.
Considering the complexity of the work, the summary, the descriptive scheme and the images with their captions, are particularly appreciated.
To make the paper even more interesting, I would like to denote three points:
- In the introduction, from line 38 (During development of the visual system, the retinal ...) it would be useful for the reader to know something more about the development of the retino-thalamic system and cortical projections, by highlighting the molecular targets involved in the growth of the structured nerves in order to further contextualize the work done on the L-lactate-GPR81 coupling. In this way the reader will understand better the meaning of the work.
- Line 41 describes how the high metabolic demands operated by neuronal development require lactate, despite a rich endowment of mitochondria for aerobic respiration. It would be appropriate to elaborate further this aspect in the text: why do mitochondria seem not to be a priority in this process?
- Finally, the discussion could describe the final meaning of your findings, whether is possible to attribute to the lactate or to the modulation of the GPR81 receptor a therapeutic and clinical value for neuroregenerative or neuroreparative purposes.
Author Response
We would like to thank the reviewer 1 for the time spent reading our manuscript and for the positive feedback provided.
In the introduction, from line 38 (During development of the visual system, the retinal ...) it would be useful for the reader to know something more about the development of the retino-thalamic system and cortical projections, by highlighting the molecular targets involved in the growth of the structured nerves in order to further contextualize the work done on the L-lactate-GPR81 coupling. In this way the reader will understand better the meaning of the work.
As requested by reviewer 1, the development of the retino-thalamic system and of the cortical projections has been succinctly described. (Lines 39- 50 in the revised version of the manuscript)
Line 41 describes how the high metabolic demands operated by neuronal development require lactate, despite a rich endowment of mitochondria for aerobic respiration. It would be appropriate to elaborate further this aspect in the text: why do mitochondria seem not to be a priority in this process?
Mitochondria are always a priority in the process of ATP production, but in high energetic demands, there is a compensation of ATP from glycolysis to fill the ATP equilibrium. The manuscript has been modified accordingly (Lines 51- 54)
Finally, the discussion could describe the final meaning of your findings, whether is possible to attribute to the lactate or to the modulation of the GPR81 receptor a therapeutic and clinical value for neuroregenerative or neuroreparative purposes.
The end of the discussion has been rewritten and the potential therapeutic implication of our research is now described. (Lines 425-433)
Reviewer 2 Report
The manuscript describes the role of L-lactate and its receptor GPR81 in the development of the mouse and hamster visual system. The work is well carried out and only minor questions remain:
- a) Please indicate whether male or female or a mix of sexes of mice/hamsters were used in the study.
- b) The statistical analysis is poorly described, it needs to describe the methods used.
- c) The authors had the options to use wild type (with presence of GPR81 protein) compared to knockout gpr81 (no GPR81 protein) mice on immunohistochemistry. Instead, they choose to depict the IHC of wild type only at E16 and in the adult mice, and the localization of GPR81 in all retinal cell types in figures 1E and 1H is not immediately evident. The same holds for the protein localizations in Figure 2A. What is true signal and what is technical artefact? And for Figure 2A, does the GPR81 protein colocalize with particular cellular compartments other than filopodia?
- d) Line 351 change “concentraions” into “concentrations”
- e) Figure S1 above the figure, change “Hoescht” into “Hoechst”
- f) Legend to Figure S1 states colocalization of GPR81 with phalloidin staining.
- g) The legend to the Western blots do not indicate the size of the expected protein bands? And does not report in the legend the TGX Stain-Free Technology for the total protein.
- h) The authors should carefully check for the correctness of “P-PKA total protein, and PKA total protein”. Is one of these figure panels wrong, since the total protein patterns should look about the same? Or are these samples run on different % gels? Or is there significant protein degradation in panel “PKA Total Protein”? Because of these unclarities, authors should present the entire Western blots for the IHC staining's (not only the selected regions) in the supplementary data.
Author Response
We sincerely appreciate the efforts of the reviewer 2 in providing constructive criticism and we addressed his/her concerns as follows:
- Please indicate whether male or female or a mix of sexes of mice/hamsters were used in the study.
Mice and hamsters of both sexes were used in this study. It is now indicated in the 2. Materials and Methods section (Lines 87-88)
- The statistical analysis is poorly described, it needs to describe the methods used.
In the 2.13 Statistical Analysis section of the manuscript, we only described how we tested the statistical distribution of our sample.
Since statistical tests change depending on the sample tested, in the revised version of the manuscript, we now indicate the specific statistical tests used in every figure legends.
- The authors had the options to use wild type (with presence of GPR81 protein) compared to knockout gpr81 (no GPR81 protein) mice on immunohistochemistry. Instead, they choose to depict the IHC of wild type only at E16 and in the adult mice, and the localization of GPR81 in all retinal cell types in figures 1E and 1H is not immediately evident. The same holds for the protein localizations in Figure 2A. What is true signal and what is technical artefact?
As suggested by reviewer #2, it could be very interesting to test the specificity of the GPR81 antibody using tissue harvested from gpr81KO mice. However, the KO used in this manuscript is not a complete KO, although the functional effects of GPR81 are abolished in these KO mice, a truncated part of the GPR81 protein containing the epitope targeted by the GPR81 antibody is still expressed.
To circumvent this obstacle and to test if the aminoacidic sequence targeted by the Sigma SAB1300790 GPR81 antibody (…fpkfyn klkicslkpk qpghsktqrp eempisnlgr rs…) was present in other proteins expressed in mice, we used the bioinformatic tool Protein BLAST (blast.ncbi.nlm.nih.gov). The only match that we obtained was the “G protein-coupled receptor 81” or the “hydroxycarboxylic acid receptor 1”, two synonyms for the GPR81. Although imperfect, this test allows us to predict that our antibody targets only the GPR81 receptor and no other proteins present in mice. This suggests a certain specificity to the anti-GPR81 used in the present study.
And for Figure 2A, does the GPR81 protein colocalize with particular cellular compartments other than filopodia?
Using fluorescent optical microscopy, it is hard to precisely identify in which growth cone’s compartment GPR81 is expressed. In results presented in the figure 2A, we observed that it is present in the axon and the growth cone, mainly in the filopodia.
- Line 351 change “concentraions” into “concentrations”
The word “concentrations” was corrected (now line 367)
- Figure S1 above the figure, change “Hoescht” into “Hoechst”
As requested, “Hoechst” was corrected
- Legend to Figure S1 states colocalization of GPR81 with phalloidin .
Figure S1 and its legend were modified accordingly to reviewer’s comment.
- The legend to the Western blots do not indicate the size of the expected protein bands? And does not report in the legend the TGX Stain-Free Technology for the total protein.
In the revised version of the Figure 4, the expecting molecular weight of each protein band was added and the TGX Stain-Free Technology was added in the figure legend.
- The authors should carefully check for the correctness of “P-PKA total protein, and PKA total protein”. Is one of these figure panels wrong, since the total protein patterns should look about the same? Or are these samples run on different % gels? Or is there significant protein degradation in panel “PKA Total Protein”? Because of these unclarities, authors should present the entire Western blots for the IHC staining's (not only the selected regions) in the supplementary data.
We want to thank reviewer 2 for his/her vigilance and we are deeply sorry for this mistake. Looking back to the original data, we realized that the wrong blot was taken for PKA total protein. The revised version of the figure 4 has been updated in the manuscript and the entire western blot for the IHC staining will be present in the supplementary .
Round 2
Reviewer 2 Report
The authors sufficiently addressed the questions.